# Importance of interface open circuit potential on aqueous hydrogenolytic reduction of benzyl alcohol over Pd/C

Guanhua Cheng[1,2], Wei Zhang[1,3], Andreas Jentys[1], Erika E. Ember[1], Oliver Y. Gutiérrez [4], Yue Liu [1,3] ✉ & Johannes A. Lercher [1,4] ✉

The open circuit potential (OCP) established by the quasi-equilibrated electrode reaction of $H_2$ and $H_3O^+_{(hydr.)}$, complicates catalytic reactions significantly. The hydrogenolysis rate of benzylic alcohol on Pd/C increases 2-3 orders of magnitude with the pH decreasing from 7 to 0.6. The reaction follows a pathway of protonated benzyl alcohol dehydration to a benzylic carbenium ion, followed by a hydride addition to form toluene. The dehydration of protonated benzyl alcohol is kinetic relevent, thus, being enhanced at lower pH. The OCP stabilizes all cationic species in the elementary steps. Particularly, the initial state (benzyl alcohol oxonium ion) is less stabilized than the dehydration transition state and the product (benzylic carbenium), thus, lowering the free energy barrier of the rate-determining step. In accordance, the rate increased with increasingly negative OCP. Beside OCP, an external negative electric potential in an electrocatlaytic system was also demonstrated to enhance the rate in the same way.

Catalyzed hydrogen additions such as hydrogenation and hydrogenolysis, are fundamental transformation for the synthesis of chemicals and energy carriers, typically catalyzed by transition metals[1–3]. The catalytic activity critically depends on the interactions between the reactive substrates and the active catalyst surface as well as on interactions with solvents and other reactive substrates. These interactions (de)stabilize ground and transition states, and therefore, influence the standard free energy barriers that determine reaction rates[4–6].

While such hydrogen addition reactions have been extensively studied and well understood at gas-solid interfaces[7,8], the presence of water induces additional complexity, as the electrochemical reactions at the interface lead to the generation of hydronium ions that may induce proton-coupled electron transfer (PCET)[9–11]. In addition, an open circuit potential (OCP) establishes spontaneously on the metal catalyst by the quasi-equilibrated electrode reaction of $H_2$ and $H_3O^+_{(hydr.)}$,

establishing an electrostatic potential gradient from the metal surface to the bulk solution. It influences the excess chemical potentials of charged and neutral species and leads to a unique self-organization of the constituents at the metal-solvent interface[12]. Thus, not only the chemisorbed species on the metal, but also the (hydrated) ions and molecules at the outer Helmholtz plane, such as $H_3O^+_{(hydr.)}$, play an important role on determining the catalytic activity.

It has been reported previously that hydronium ions greatly enhance reaction rates catalyzed by Pt and Pt-group metals, by enabling an energetically favored PCET pathway[13–15] and by weakening the hydrogen binding standard free energy[16]. This points to the more general impact of the proton activity at the electrode surface plays on the thermodynamic state of the reacting substrates and, hence, on the catalytic pathways[17,18].

On a first view, the reductive one-step conversion of the alcohol C–O bond to a C-H bond and water resembles the hydrogenolytic

[1]Technische Universität München, Department of Chemistry and Catalysis Research Center, Lichtenbergstraβe 4, Garching D-85748, Germany. [2]Key Laboratory for Liquid-Solid Structural Evolution and Processing of Materials (Ministry of Education), School of Materials Science and Engineering, Shandong University, Jingshi Road 17923, Jinan 250061, PR China. [3]Shanghai Key Laboratory of Green Chemistry and Chemical Processes, School of Chemistry and Molecular Engineering, East China Normal University, Shanghai, PR China. [4]Institute for Integrated Catalysis, Pacific Northwest National Laboratory, 902 Battelle Boulevard, Richland, WA 99352, USA. ✉e-mail: liuyue@chem.ecnu.edu.cn; johannes.lercher@ch.tum.de

cleavage of C-C bonds. It has been reported that the activity and selectivity for this C−O bond cleavage is sensitive to the acidity of the catalyst support or the acidity of solvents[19–24], and that neutralizing the acid sites will eliminate the enhancement[22,25]. Extensive mechanistic studies for benzylic alcohol hydrogenolysis, led to the formulation of two main reaction mechanisms proposed, i.e., one hypothesizing that the C−O bond scission proceeds via a "bifunctional" dehydration-hydrogenation route[19,26], and another via a direct C-OH bond cleavage by hydrogen insertion[19,22,27,28]. Thus, the acid functionality either catalyzes dehydration or converts the carbonyl to a better leaving group. However, the elementary steps (which determine the catalytic pathway) involving $H_3O^+_{(hydr.)}$ and the metal in this conversion and in particular the role of the ionic environment at the interface are not understood and prevent formulating a generally accepted reaction mechanism.

Therefore, we address in this contribution the role of the OCP and hydronium ions at the metal surface and outer Helmholtz plane for benzyl alcohol hydrogenolysis on Pd/C in an aqueous environment. Pd has been chosen because of its high activity and selectivity towards C−O bond cleavage[29]. The pathway identified shows that the reaction proceeds via the protonation of the OH group making it a better leaving group ($H_2O$), leading to the formation of a benzylic carbocation, and yielding toluene after hydride addition (in contrast to the proton elimination in conventional dehydration). Using kinetic analysis and demonstrating kinetic isotope effects (KIE), we show that the dehydration of the protonated alcohol is the rate-determining step. The hydronium ions affect the reaction by changing the electric potential on Pd as well as the transition state concentration.

## Results and discussion
### Impact of the hydronium ion activity on benzyl alcohol hydrogenolysis

$$C_6H_5CH_2OH \xrightarrow{H_2/H_3O^+,Pd} C_6H_5CH_3 + H_2O \qquad (Rxn1)$$

Reductive elimination of water from benzyl alcohol on Pd/C involves cleavage of the C−O bond of benzyl alcohol to form toluene and water (Rxn 1). Products of aromatic ring hydrogenation, such as cyclohexylmethanol and methylcyclohexane, were not observed. The turnover frequency (TOF) decreased with increasing pH (Fig. 1a) and was very low above pH of 5, independently of the type of buffer solution used. This decrease suggests a critical role of hydronium ions for the catalytic conversion. In the absence of Pd, benzyl alcohol was not converted in acidic media (pH 1), demonstrating that the reaction is metal catalyzed. The reaction order in hydronium ions varies with the pH range from 0.6 to 7 (Fig. 1b).

Considering that hydronium ions are involved in the reaction, let us hypothesize that the elementary steps occur in the space ranging from the Pd surface itself to the outer Helmholtz plane. In this case the rate equation for the reaction ($r_{BA}$, subscript BA denoting benzyl alcohol) is:

$$r_{BA} = -\frac{dC_{BA}}{dt} = k_{eff}C_{BA}^{\alpha}P_{H_2}^{\beta}a_{H^+}^{\gamma} \qquad (1)$$

$k_{eff}$ is the effective rate constant; $a_{H^+}$ is the hydronium ion activity in the bulk phase, $\alpha$, $\beta$ and $\gamma$ are the apparent reaction orders with respect to benzaldehyde, $H_2$ and $H_3O^+_{(hydr.)}$.

The reaction orders for both benzyl alcohol and $H_2$ were measured at pH 2.5 and pH 5 (Fig. S1 and Table 1). At pH 2.5, the reaction order in benzyl alcohol is slightly positive ($\alpha = 0.08$, Fig. S1a) and near 0th order with respect to $H_2$ ($\beta = 0.09$, Fig. S1b). At pH 5, the reaction order with respect to both benzyl alcohol and $H_2$ are near 0 (Figs. S1c and S1d). The reaction order being 0th or close to 0th for organic substrate is common in hydrogenation and hydrogenolysis reactions[28,30–32], typically interpreted as a result of the high coverage of substrate. The reaction order of 0th for $H_2$ indicates that either H coverage on Pd surface is also very high, i.e., near saturated coverage under noncompetitive adsorption with organic substrates or the adsorption sites are half occupied by both the substrates and H under competitive adsorption, or that the adsorbed H does not participate in the rate determining step.

In order to estimate whether the Pd surface has a substantial coverage of H, cyclic voltammetry on Pd/C (20 mg, 30 wt.%) with different concentrations of benzyl alcohol was measured at pH 2.5 (Fig. 2a) and pH 5 (Fig. 2b). In presence of benzyl alcohol, the peak of underpotentially deposited hydrogen was reduced ($H_{upd}$, 0.06−0.34 V vs. RHE at pH 2.5 and 0.09−0.43 V vs. RHE at pH 5), and totally disappeared at a concentration of 500 μM (Fig. 2a, b). Comparing the $H_{upd}$ current in presence and absence of benzyl alcohol allows to determine the fraction of H that was inhibited (blocked) by benzyl alcohol (Fig. 2c, d). The drastically decreasing $H_{upd}$ with increasing benzyl alcohol concentration indicates that benzyl alcohol binds much stronger than H on Pd, which will lead to a full coverage of the Pd surface by benzyl alcohol and a very low coverage with H under our reaction condition (benzyl alcohol concentration between 6 and 80 mM). Therefore, the possibility of a high coverage of H is excluded, and the observed near zero reaction order of $H_2$ in benzyl alcohol reaction indicates that the H adatoms are only involved in elementary steps after the rate determining step.

If this is the case, benzyl alcohol conversion should only have a small kinetic isotope effect for gaseous $H_2$ vs. $D_2$ (as shown indeed in Fig. 3). The conversion rate of benzyl alcohol has only small differences for reactions with $H_2$ and $D_2$ in either $H_2O$ (rate ratio of 1.1) or $D_2O$ (rate ratio of 1.0) at 298 K and 1 bar $H_2$ or $D_2$. In comparison, the ratio of the rate in $H_2O$ to that in $D_2O$ is 1.4 in $H_2$ and 1.3 in $D_2$. This sensitivity indicates that H or D from water or $H_3O^+_{(hydr.)}$ are involved in the kinetically relevant steps.

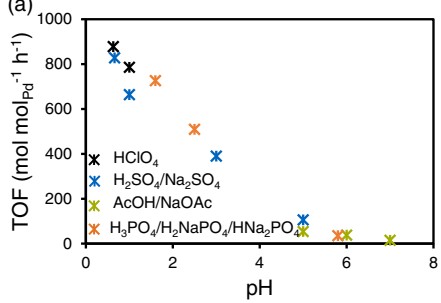
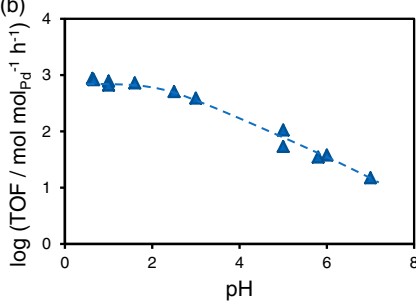

**Fig. 1 | Influence of pH on the activity of benzyl alcohol hydrogenolysis. a** TOF as a function of pH in the catalytic conversion of benzyl alcohol to toluene on Pd/C. **b** Log TOF as a function of pH to obtain the apparent reaction order in bulk hydronium ion with a dashed line as a guide to the eye. Reaction condition: 0.2 M buffer solution, 298 K and 1 bar $H_2$. The legend indicates different buffer compositions.

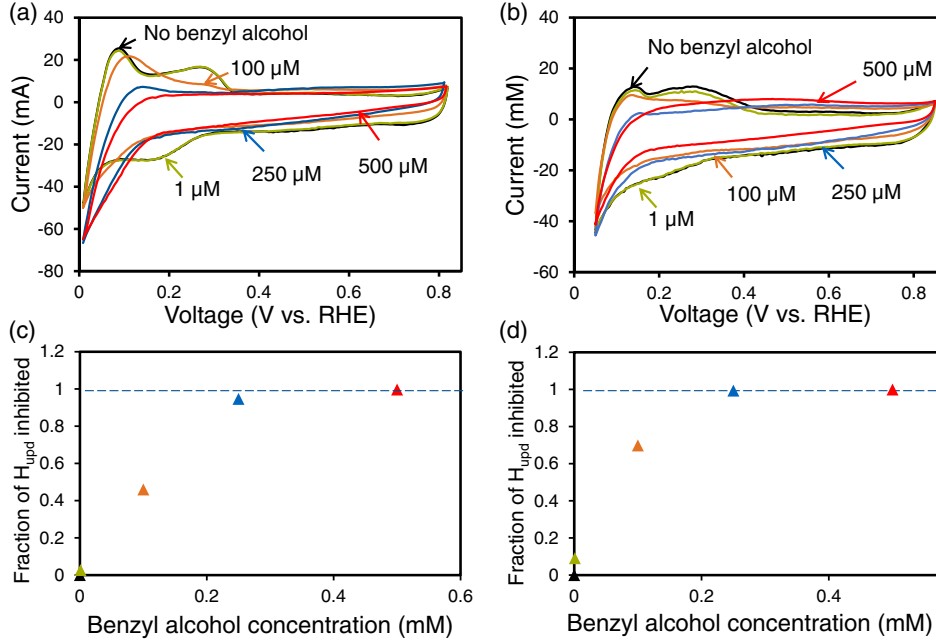

**Fig. 2 | Influence of the benzyl alcohol concentration on the coverage of H on Pd/C.** Cyclic voltammograms at a scan rate of 5 mV s$^{-1}$ showing hydrogen underpotential deposition on Pd/C (30 wt.%) at room temperature with benzyl alcohol concentration varying from 0 to 500 μM and at **a** pH 2.5 in phosphate buffer solution and **b** pH 5 in acetate buffer solution. The calculated fraction of underpotentially deposited hydrogen that is inhibited on Pd vs. benzyl alcohol concentration **c** pH 2.5 in phosphate buffer solution and **d** pH 5 in acetate buffer solution, which is calculated from the decrease in H$_{upd}$ charge relative to that without benzyl alcohol obtained from (**a**) and (**b**), respectively.

## Possible reaction pathways

Three pathways to convert benzyl alcohol to toluene are conceivable, i.e., (i) direct hydrogenolysis (Fig. 4a), (ii) partial hydrogenation–dehydration–re-hydrogenation ((Fig. 4b),) and (iii) protonation–dehydration–hydride addition (Fig. 4c).

In the first pathway, benzyl alcohol adsorbs on Pd molecularly (BA$_{ad}$); dissociatively adsorbed H$_2$ acts as the reducing agent (H$_{ad}$). Then, following an SN$_2$ mechanism, H$_{ad}$ attacks the benzylic carbon and displaces the hydroxyl group, leading to the formation of toluene[28]. This pathway has hydrogen involved in all relevant cleavage steps and is incompatible with the absence of an isotope effect.

In the second reaction pathway, benzyl alcohol and H$_2$ are adsorbed and the aromatic ring of benzyl alcohol is partial hydrogenated providing a β-H. The intramolecular dehydration yields the olefin product, followed by hydrogen addition leading to toluene and/or hydrogenation to 5-methylcyclohexa-1,3-diene. This mechanism is similar to that reported for C–O bond cleavage of 1-(4-isobutylphenyl)ethanol over Pd supported on acidic carbon[19]. As benzyl alcohol does not have a hydrogen in the β position for dehydration, the aromatic ring needs to be partly hydrogenated, in analogy to the reductive solvolysis of aryl ethers[3,33]. This reaction pathway is also excluded, because 5-methylcyclohexa-1,3-diene was not detected, and deuterium was not observed in the aromatic ring for reactions performed in D$_2$ and D$_2$O (SI, Note S1).

Thus, the third pathway, water elimination upon protonation of the hydroxyl group to a carbocation in presence of hydronium ions is

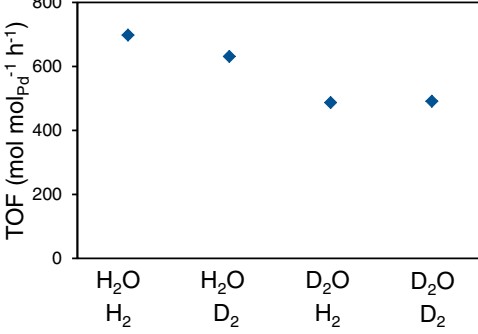

**Fig. 3 | Turnover rates of benzyl alcohol hydrogenolysis in H$_2$O·H$_2$, H$_2$O·D$_2$, D$_2$O·H$_2$, and D$_2$O·D$_2$.** Reaction condition: at 298 K and atmospheric pressure with a benzyl alcohol concentration of 20 mM on Pd/C and 0.2 M phosphoric acid.

concluded to be the most likely reaction pathway and will be further analyzed in detail.

## Factors influencing benzyl alcohol hydrogenolysis

The elementary steps of the pathway are shown in Fig. 4c. It should be emphasized that the presence of water, H$_2$, and H$_3$O$^+$$_{(hydr.)}$ will lead to establishing the corresponding OCP at the surface, which enables quasi-equilibrated interconversion of the species involving electrons of Pd.

The electrochemical quasi-equilibrium (hydrogen electrode reaction) will be established between all species on the metal surface establishing the OCP, as shown in the reaction equation Rxn 2 (Step g in Fig. 4c).

$$H^* + H_2O \rightleftharpoons H_3O^+ + e^- + * \tag{Rxn2}$$

The catalytic cycle starts with benzyl alcohol adsorption on vacant Pd sites (*) (Step a), afterwards, the adsorbed benzyl alcohol (BA*) is

## Table 1 | Summary of reaction rates and reaction orders

| pH | TOF (mol mol$_{Pd}$$^{-1}$h$^{-1}$)[a] | Reaction order benzyl alcohol (α) | Reaction order H$_2$ (β) |
|---|---|---|---|
| 2.5 | 527 | 0.08 | 0.09 |
| 5 | 162 | −0.01 | 0.09 |

[a]TOF at 1 bar H$_2$ and 298 K with a benzyl alcohol concentration of 20 mM.

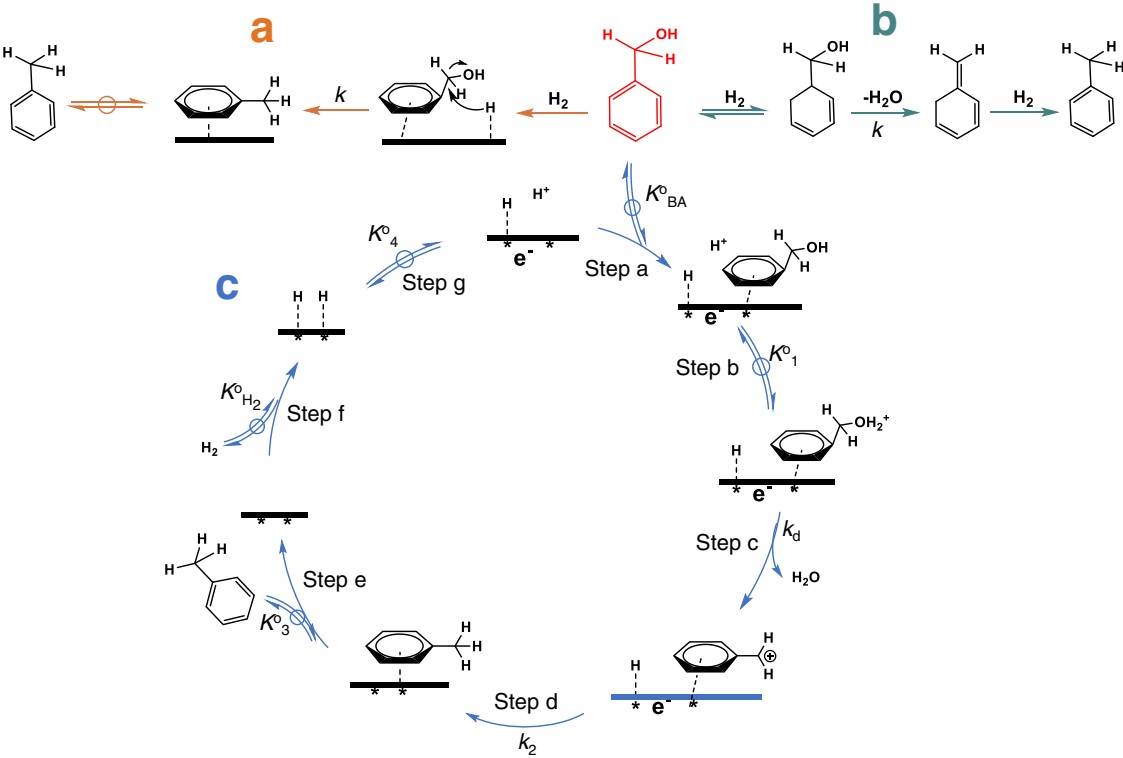

**Fig. 4 | Reaction pathway of benzyl alcohol hydrogenolysis. a** Direct hydrogenolysis, **b** partial hydrogenation– dehydration–re-hydrogenation and **c** protonation– dehydration–hydride addition (Fig. 4**c**). Elementary steps in Fig. 4**a**, **b** are omitted and the hydronium ion is expressed as proton in the scheme.

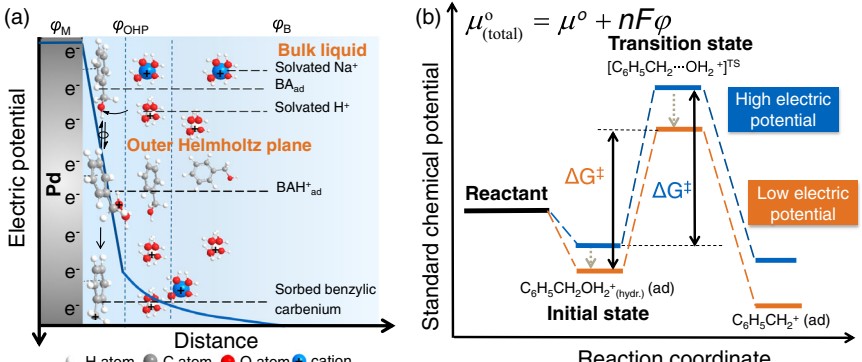

**Fig. 5 | Schematic illustration of the rate-determining step and the energy diagram. a** The Reaction steps of benzyl alcohol hydrogenolysis between Pd surface and Helmholtz plane and the potential profile as a function of distance from Pd surface. $\varphi_M$, $\varphi_{OHP}$, $\varphi_B$ are potentials on Pd surface, at outer Helmholtz plane and in the bulk, respectively. **b** Standard chemical potential profiles of the initial state and transition state under low and high electric potentials.

protonated (Step b, $C_6H_5CH_2OH_2^{+*}$, BAH$^{+*}$), followed by elimination of water and the formation of the benzyl carbocation ($C_6H_5CH_2^{+*}$) (Step c). Toluene is then formed via hydride addition (Step d) and desorption from the Pd surface (Step e). $H_2$ dissociatively adsorbs on a Pd site pair (*-*), forming two adsorbed H atoms (H*) (Step f). The adsorption-desorption steps (Step a, e and f) are generally much faster and are, hence, considered to be quasi-equilibrated. Hydride addition (Step d) is not kinetic relevant, as a substantial KIE has not been observed when replacing $H_2$ by $D_2$. Therefore, the dehydration step (Step c) is considered to be rate-determining, when deriving the rate equation.

In aqueous phase, the OCP on the Pd surface induces an electrical double layer (EDL) at the Pd-water interface, where the Pd-catalyzed benzyl alcohol reduction occurs. Figure 5a shows the reaction elementary steps (Step b–c) in the space between the metal surface and

outer Helmholtz plane (OHP). The chemisorbed species with their directed bonding to the surface, are located in the inner Helmholtz plane (IHP), while the nonspecifically adsorbed ions like hydrated hydronium ions and hydrated metal cations are located at the outer Helmholtz plane. The electric potential on Pd surface is determined by the equilibrium between $H_2$ and hydronium ion (Rxn 2). Under quasi-equilibrium conditions the OCP, i.e., the electrode potential of Pd, is expressed by the Nernst Equation (Eq. 2),

$$\varphi_M = \varphi_{SHE} + \frac{RT}{F} \ln \frac{a_{H^+}}{\sqrt{P_{H_2}}} \qquad (2)$$

in which $\varphi_{SHE}$ is standard hydrogen electrode potential (SHE) (defined as 0 V for convenience). The $a_{H^+}$ is the hydronium ion activity in the bulk solution. It should be noted that $\varphi_M$ is the electric potential

difference between Pd surface and electrolyte. The electric potential decays with distance from Pd surface until reaching the level of the bulk solution, which is considered as 0 V (Fig. 5a).

At the Pd surface, adsorbed H ($H_{ad}$), adsorbed benzyl alcohol ($BA_{ad}$), protonated benzyl alcohol ($BAH^+_{ad}$) and unoccupied sites (*) coexist. During a progressing reaction, a very small concentration of molecules in the transition state (TS) of the rate determining step must also exist (Fig. 4, step c). The expressions of chemical potentials or electrochemical potentials for all the species are

$$\mu_{BA_{aq}} = \mu^o_{BA_{aq}} + RT\ln a_{BA_{aq}} \tag{3}$$

$$\mu_{BA_{ad}} = \mu^o_{BA_{ad}} + RT\ln\frac{\theta_{BA_{ad}}}{\theta^*} \tag{4}$$

$$\mu_{H_2} = \mu^o_{H_2} + RT\ln P_{H_2} \tag{5}$$

$$\mu_{H_{ad}} = \mu^o_{H_{ad}} + RT\ln\frac{\theta_{H_{ad}}}{\theta^*} \tag{6}$$

$$\mu_{BAH^+_{ad}} = \mu^o_{BAH^+_{ad}} + RT\ln\frac{\theta_{BAH^+_{ad}}}{\theta^*} + F\varphi_{BAH^+_{ad}} \tag{7}$$

$$\mu_{H^+} = \mu^o_{H^+} + RT\ln a_{H^+} + F\varphi_B \tag{8}$$

$$\mu_{TS} = \mu^o_{TS} + RT\ln\frac{\theta_{TS}}{\theta^*} + F\varphi_{TS} \tag{9}$$

The $\mu_{BAaq}$, $\mu_{BAad}$, $\mu_{H_2}$, $\mu_{Had}$, $\mu_{H^+}$, $\mu_{BAH^+ad}$ and $\mu_{TS}$ are chemical or electrochemical potential of benzyl alcohol in bulk, sorbed benzyl alcohol, gas $H_2$, sorbed H, hydronium ion in bulk, sorbed protonated benzyl alcohol and transition state, respectively. The $\mu^o_{BAaq}$, $\mu^o_{BAad}$, $\mu^o_{H_2}$, $\mu^o_{Had}$, $\mu^o_{H^+}$, $\mu^o_{BAH^+ad}$ and $\mu^o_{TS}$ are their corresponding chemical/electrochemical potential at standard state, defined at 1 bar $H_2$, 1 M concentration and a coverage of 0.5, respectively. The transition state (TS) and sorbed protonated benzyl alcohol ($BAH^+_{ad}$) are positively charged species, thus, they are considered to be located closer to the outer Helmholtz plane (Fig. 5a). The $\varphi_{BAH^+ad}$, $\varphi_B$ and $\varphi_{TS}$ are the electric potentials at the position of $BAH^+_{ad}$, bulk solution and transition state, respectively. Generally, $\varphi_B$ is denoted as 0 V. It should be noted that the electrochemical potentials of the charged species $BAH^+_{ad}$, $H_3O^+_{(hydr.)}$ and TS are affected not only by the activity, but also by the electric potential at their located position. Figure 5b illustrates, how the electric potentials affect the chemical potential (free energy) of the charged reacting species. A more negative (low) electric potential stabilizes the cationic species, i.e., the initial state (benzyl alcohol oxonium ion), its dehydration transition state and the product (benzylic carbenium). The extent of stabilization depends on the location of these species in the EDL. Considering the hydrogen bonding at the outer Helmholtz layer, the transition state at a closer location to the Pd surface leads to a higher stabilization as it has less hydrogen bonding with water compared to the initial state, that consequently decreases their energy differences and the intrinsic energy barrier.

Because the dehydration step (Step c) is considered to be the RDS, the adsorption of substrate (Step a) and $H_2$ (Step f) as well as the protonation of benzyl alcohol (Step b) are considered to be quasi-equilibrated. This leads to several relations of chemical potentials,

$$\mu_{BA_{aq}} = \mu_{BA_{ad}} \tag{10}$$

$$\mu_{H_2} = 2\mu_{H_{ad}} \tag{11}$$

$$\mu_{BA_{ad}} + \mu_{H^+} = \mu_{BAH^+_{ad}} = \mu_{TS} \tag{12}$$

With these equations, the reaction rate is derived on basis of transition state theory as (derivation details in SI):

$$r = \frac{k_B T}{h} \frac{\exp\left(\frac{\mu_{BA_{aq}} + \mu_{H^+} - \mu^o_{TS} - F\varphi_{TS}}{RT}\right)}{1 + \exp\left(\frac{1/2\mu_{H_2} - \mu^o_{H_{ad}}}{RT}\right) + \exp\left(\frac{\mu_{BA_{aq}} - \mu^o_{BA_{ad}}}{RT}\right) + \exp\left(\frac{\mu_{BA_{aq}} + \mu_{H^+} - \mu^o_{BAH^+_{ad}} - F\varphi_{BAH^+_{ad}}}{RT}\right)} \tag{13}$$

Thus, the reaction rate (Eq. 13) is affected by the chemical potentials of aqueous benzyl alcohol ($\mu_{BAaq}$), hydronium ion ($\mu_{H^+}$) and $H_2$ ($\mu_{H_2}$) as well as by $\varphi_{TS}$ and $\varphi_{BAH^+ad}$. The chemical potentials are dependent on the benzyl alcohol concentration, $H_3O^+_{(hydr.)}$ concentration and $H_2$ pressure, respectively. To gauge the influence of each parameter influences the reaction, we introduce the partial derivative of the reaction rate to this parameter.

$$\frac{\partial \ln r}{\partial \ln a_{BA_{aq}}} = RT\frac{\partial \ln r}{\partial \mu_{BA_{aq}}} = 1 - \theta_{BA_{ad}} - \theta_{BAH^+_{ad}} \tag{14}$$

$$\frac{\partial \ln r}{\partial \ln P_{H_2}} = -\frac{1}{2}\theta_{H_{ad}} + \frac{\partial \varphi_{TS}}{2\partial \varphi_M} - \theta_{BAH^+_{ad}}\frac{\partial \varphi_{BAH^+_{ad}}}{2\partial \varphi_M} \tag{15}$$

$$\frac{\partial \ln r}{\partial \ln a_{H^+}} = 1 - \theta_{BAH^+_{ad}} - \frac{\partial \varphi_{TS}}{\partial \varphi_M} + \theta_{BAH^+_{ad}}\frac{\partial \varphi_{BAH^+_{ad}}}{\partial \varphi_M} \tag{16}$$

$$\frac{\partial \ln r}{\partial \varphi_M} = \frac{F}{RT}\left(-\frac{\partial \varphi_{TS}}{\partial \varphi_M} + \theta_{BAH^+_{ad}}\frac{\partial \varphi_{BAH^+_{ad}}}{\partial \varphi_M}\right) \tag{17}$$

Equations 14, 15 and 16 represent the reaction orders with respect to benzyl alcohol, $H_2$ and $H_3O^+_{(hydr.)}$. It is noticeable that their reaction orders are functions of the coverages of corresponding species. In particular, the reaction orders of $H_2$ and $H_3O^+_{(hydr.)}$ contain the term $\varphi_M$ (Eqs. 15 and 16), indicating that they can influence the reaction rate by changing the electric potential on Pd because they are involved in the hydrogen electrode reaction, establishing the OCP. On the other hand, benzyl alcohol does not affect the electric potential, so its reaction order expression does not contain $\varphi_M$ (Eq. 14). The measured reaction order of benzyl alcohol was 0, indicating the $(\theta_{BA} + \theta_{BAH^+_{ad}})$ being 1 according to Eq. 14, and meaning Pd surface is fully saturated by benzyl alcohol and its protonated form $BAH^+$. This agrees perfectly with the CV measurements (Fig. 2) that Pd surface is saturated with benzyl alcohol and derived species.

The influence of electric potential $\varphi_M$ on the rate is expressed in Eq. 17. To verify this, Pd/C was loaded on an electrode and exposed to a negative electric potential (vs. SHE). Figure 6 shows the conversion rate of benzyl alcohol to toluene at pH 1.6 under different electric potentials. A larger reaction rate was observed under a more negative electric potential. Stabilizing the carbenium ion product after dehydration, is concluded to lead to a lower transition state following the Polanyi relation[34]. We assume that transition state is closer to Pd surface compared to protonated benzyl alcohol ($BAH^+$), as it will be then less hydrogen bonded in the aqueous environment.

It is unclear whether $H_2$ competes for the active sites on Pd or whether it adsorbs in between adsorbed benzyl alcohol molecules[31]. Under the conditions studied this effect is negligible due to the small coverage of $H_{ad}$. Therefore, we conclude that the main effect of $H_2$ on the reaction rate is related to its influence via $\varphi_M$ (Eq. 15). A higher $H_2$ pressure is equivalent to a more negative $\varphi_M$ (Eq. 2).

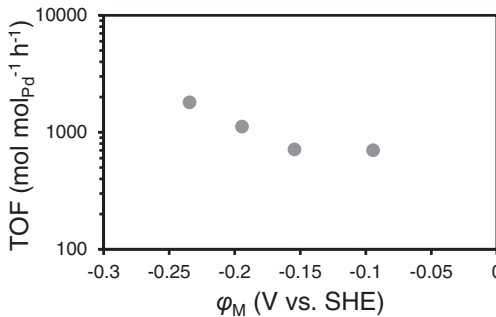

**Fig. 6 | TOF as a function of electric potential of Pd at pH 1.6.** Reaction condition: 10 mg Pd/C (5wt.%) at 298 K.

The $H_3O^+_{(hydr.)}$ concentration is directly related to the concentration of $BAH^+_{ad}$, and the reaction order in $H_3O^+_{(hydr.)}$ can partially reflect the relative abundance of $BA_{ad}$ and $BAH^+_{ad}$ on Pd surface (the none electric potential related part in Eq. 16). This agrees with the changing reaction order in $H_3O^+_{(hydr.)}$ during the whole pH range from pH 0.6 to pH 7 (Fig. 1b) and the fact that a smaller reaction order was obtained at a lower pH. However, increasing $H_3O^+_{(hydr.)}$ concentration also increases $\varphi_M$, which inhibits the reaction. As shown in Fig. 7, for each pH, the reaction rate increases with $H_2$ pressure, caused by the decreasing of electric potential at either OCP or under external electric potentials. In comparison, to decrease pH can be resolved into two components: $BAH^+_{ad}$ coverage (or transition state concentration) increasing and $\varphi_M$ increasing. Apparently, the increase in $BAH^+_{ad}$ coverage overcomes the negative influence by the increase of the electric potential, and leads to the increased reaction rate with decreasing pH.

It is shown that the reductive elimination of benzyl alcohol catalyzed by Pd/C is highly sensitive to the quasi-equilibrated electrochemical steps at the metal-water interface. The conversion rate of benzyl alcohol hydrogenolysis increased by 2–3 orders of magnitude with decreasing the pH from 7 to 0.6. Kinetic analysis and isotope-labeling study show that the reaction follows a pathway of protonating benzyl alcohol at the hydroxyl group, followed by elimination of water to form a benzyl carbenium ion, and hydride addition to the benzyl carbenium ion to form toluene. The elimination to form the benzyl carbenium ion is concluded to determine the overall reaction rate.

Acting as proton donor, a high $H_3O^+_{(hydr.)}$ activity increases the reaction rate by increasing the concentration of the reacting initial state ($BAH^+_{ad}$) due to the higher probability of protonation. Besides this conventional effect, $H_3O^+_{(hydr.)}$ also influences the OCP. The OCP is the electric potential established by the quasi-equilibrated electrode reaction of $H_2$ and $H_3O^+_{(hydr.)}$ at the metal-water interface; it decreases with $H_2$ pressure (negative correlation) and increases with the activity of $H_3O^+_{(hydr.)}$ (positive correlation). As positively charged species, both the reacting initial state ($BAH^+_{ad}$) and its dehydration transition state are stabilized by the negative electric potential at the metal-water interface. Because of the strong hydrogen bonding at the outer Helmholtz layer, the benzyl carbenium ion intermediate is closer to the metal surface, as it has less hydrogen bonding with water compared to the initial state ($BAH^+_{ad}$). Thus, the negative charge at the metal surface provides relative more stabilization to the benzyl carbenium ion, that consequently decreases their energy differences and the reaction energy barrier. Such effect is weakened at higher activities of $H_3O^+_{(hydr.)}$ because of the less negative OCP.

By influencing OCP, $H_2$ also influences the conversion rate, although $H_2$ as a reactant is only involved in the steps after rate determining step. A high pressure of $H_2$ corresponds to a more negative OCP, resulting in a lower activation energy and higher reaction rate. The results highlight the complex interplay that needs to be considered, when interpreting reactions at the water-metal interface.

More general, it underlines the importance of OCP at the solid-aqueous interface for the reactions involving charged reactants, intermediates and transition states.

## Methods

### Chemicals and catalyst

Pd/C catalyst with a Pd content of 5 wt.% was purchased from Sigma Aldrich. The Pd particle size is 2.9 nm on average measured by TEM, and metal dispersion is 33% measured by $H_2$ adsorption. All chemicals were obtained from commercial suppliers and used as received, including benzyl alcohol (Sigma-Aldrich, ≥99.0%) and chemicals for buffer solutions (Sigma-Aldrich, ≥99.9%, e.g., $HClO_4$, $H_3PO_4$, $NaH_2PO_4$, $Na_2HPO_4$, $H_2SO_4$, $Na_2SO_4$, $CH_3COOH$, $CH_3COONa$), NaCl (Sigma-Aldrich, ≥99.9%), ethyl acetate (Sigma-Aldrich, ≥99.9%, HPLC), and 2-cyclohexen-1-one (Sigma Aldrich, >99%). High purity water, treated with a Milli-Q water purification system until a resistivity of 18.2 MΩ cm, was used in all experiments. $H_2$ (Air Liquide, >99.99%) was used for hydrogenation.

### Catalyst characterization

The specific surface area of the catalyst was determined (according to BET) from $N_2$ physisorption, which were measured at 77 K on a PMI automated BET sorptometer. The samples were first outgassed at 523 K in vacuum (<0.001 mbar) for 20 h before measurement.

The dispersion of the metals was determined by $H_2$ chemisorption on Thermo Scientific Surfer Analyzer. The Pd/C catalysts were treated in vacuum at 588 K for 1 h and then cooled to 313 K. A first set of $H_2$ adsorption isotherm was measured from 1 to 40 kPa. Afterwards, the samples were outgassed at the same temperature for 1 h to remove the physisorbed $H_2$, followed by a second set of isotherms being measured, which corresponded to physisorbed $H_2$. The difference of the two isotherms was the chemisorbed hydrogen on Pd. The concentrations of surface Pd atoms were determined by extrapolating the saturated region of the difference isotherms to zero hydrogen pressure and using the value as the number of surface Pd atoms assuming a stoichiometry of one hydrogen to one Pd atom. Then the dispersion of Pd was calculated by comparing the surface Pd atoms to the total Pd atoms.

The size of Pd particle was determined by transmission electron microscopy (TEM, JEOL JEM-2011) with an accelerating voltage of 120 keV. Statistical treatment of the metal particle size was done by counting at least 300 particles detected in several places of the grid. The Pd/C used in this work is the same as in our previous work and the TEM image[4].

### Catalytic reaction measurements under OCP (open circuit potential)

Benzyl alcohol hydrogenolysis was carried out in a batch reactor with 5–10 mg Pd/C (5 wt.% Pd loading) in 0.2 M buffer solution. Typical measurements under atmospheric pressure were performed with $H_2$ or diluted $H_2$ using $N_2$ (flow rate of 10 mL min$^{-1}$) flowing through the reactant solution at 296 K and 600 rpm. Reactions at higher $H_2$ pressures were performed in a 300 mL Hastelloy PARR reactor, and air was removed from the reactor by introducing 20 bar $H_2$, followed by depressurizing the reactor for three times. The benzyl alcohol concentration was 20 mM. After the reaction, the solution was extracted using ethyl acetate with 2-cyclohexen-1-one as the internal standard for quantification. NaCl was used to increase the extraction efficiency. The extracted ethyl acetate was treated with $Na_2SO_4$ to remove dissolved water. Quantitative analyses of the samples were performed by gas chromatography equipped with a Wax capillary column (30 m × 250 μm) and a flame ionization detector (FID). Reaction orders of $H_2$ were determined by changing $H_2$ pressure from 0.2 bar to 20 bar.

The reactions in $D_2O$ or/and $D_2$ were carried out in a 100 mL Hastelloy PARR reactor. 20 mM benzyl alcohol, 30 mL 0.2 M $D_3PO_4$

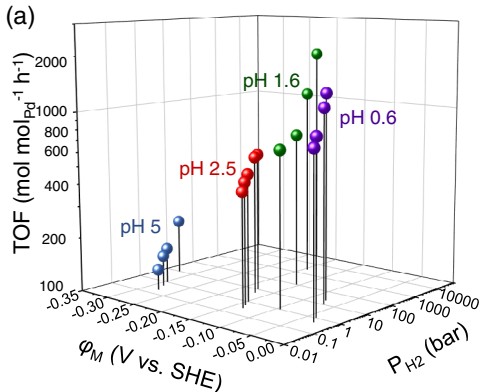

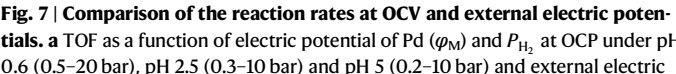

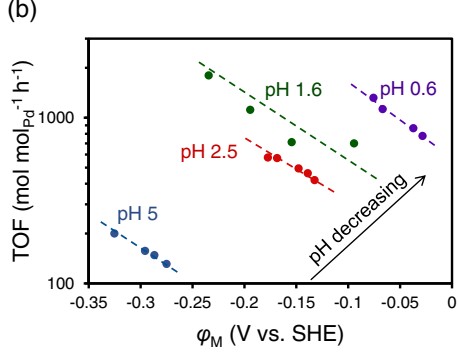

**Fig. 7 | Comparison of the reaction rates at OCV and external electric potentials. a** TOF as a function of electric potential of Pd ($\varphi_M$) and $P_{H_2}$ at OCP under pH 0.6 (0.5–20 bar), pH 2.5 (0.3–10 bar) and pH 5 (0.2–10 bar) and external electric potential at pH 1.6 on Pd/C (5wt.%) at 298 K. **b** The corresponding projection onto the plane of TOF vs. $\varphi_M$. A hypothetical $P_{H_2}$ is used for pH 1.6 that is equilibrated with the externally added electric potentials.

solution in $D_2O$ (99.8%, Sigma), 1 bar $D_2$ and 10 mg Pd/C were sealed in the reactor for reaction at 296 K and 750 rmp. Before the reaction, air was removed from the reactor by introducing 20 bar $D_2$, followed by depressurizing the reactor for three times. After the experiment, ethyl acetate was used to extract the chemical species and then were quantified using GC-MS equipped with a HP-5 capillary column.

### Electrocatalytic reaction

Experiments were carried out in a H cell with a Nafion 117 proton exchange membrane (Ion Power, Inc.) to separate the cathodic and anodic compartments. An electrochemical workstation VSP-300, Bio Logic was used to perform electrochemical procedures. A piece of carbon felt (Alfa Aesar >99.0%, 3.2 mm thickness) infiltrated with 10 mg Pd/C connected to a graphite rod (Sigma Aldrich, 99.99%) was used as working electrode in the cathode compartment. A platinum wire (Alfa Aesar, 99.9%) was used as counter electrode in the anodic compartment. Ag/AgCl was used as a reference electrode. The cathode and anode compartments were filled with 60 mL 0.2 M $H_3PO_4$ solution as electrolyte solution. All reactions were performed at atmospheric pressure at constant potential referred to the reverse hydrogen electrode (RHE). The catalyst was first activated under a constant current of −40 mA for 10 min before adding benzyl alcohol (20 mM) into the cathode compartment. Then an electric potential was applied for reaction. Product analysis is the same as performed in the catalytic reaction under OCP.

### Cyclic voltammetry

The coverage of benzyl alcohol on Pd surface was determined via cyclic voltammetry in a Teflon H-cell with a Nafion 117 proton exchange membrane to separate cathodic and anodic compartments. 20 mg Pd/C (30 wt%, Sigma Aldrich) on carbon felt (Alfa Aesar >99.0%, 3.2 mm thickness) was used as working electrode, a Ag/AgCl electrode with double-junction was used as reference electrode, and a platinum wire (Alfa Aesar, 99.9 %) was used as counter electrode. The electrolytes are phosphate buffer (pH 2.5) and acetate buffer (pH 5). The cyclic voltammetry was measured at 5 mV s$^{-1}$ with 20 mL min$^{-1}$ $N_2$ bubbling, and under different concentrations of benzyl alcohol. The sites blocked by benzyl alcohol (benzyl alcohol coverage) was calculated based on the area difference of the underpotentially deposited hydrogen ($H_{upd}$) peak before and after the benzyl alcohol addition[35]. The same measurement was also performed on Pd/C (5 wt.%), but it was difficult to distinguish the $H_{upd}$ peak from the double layer capacitance background current induced by the carbon support.

## Data availability

All data are available within the paper and its Supplementary Information files and Source Data file. Source data are provided with this paper.

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

## Acknowledgements

G.C. is grateful to the Chinese Scholarship Council for the financial support. J.A.L. and O.Y.G. acknowledge the support by the U.S. Department of Energy (DOE), Office of Science, Office of Basic Energy Sciences (BES), Division of Chemical Sciences, Geosciences and Bios-ciences (Impact of catalytically active centers and their environment on rates and thermodynamic states along reaction paths, FWP 47319). Y.L. and W.Z. acknowledge the support by the Open Project Program of Academician and Expert Workstation, Shanghai Curui Low-Carbon Energy Technology Co., Ltd.

## Author contributions

G.C. carried out the reactions and performed the characterizations of catalysts. W.Z., A.J., E.E.E., and O.Y.G. cooperated with the discussion and provided valuable suggestions. Y.L. and J.A.L. supervised the work and provided guidance throughout the project. The manuscript was written through contributions of all authors. All authors have given approval to the final version of the manuscript.

## Funding

## Competing interests

The authors declare no competing interests.
