## [Peer Review File · Nature Communications]

Importance of Interface Open Circuit Potential on Aqueous Hydrogenolytic Reduction of Benzyl Alcohol over Pd/CREVIEWER COMMENTS

Reviewer #1 (Remarks to the Author):

This manuscript provides insight into the role of hydronium ions and an open circuit potential on the hydrogenolysis of benzyl alcohol using Pd/C. This leads to the conclusion that the dehydration of the protonated alcohol is the rate-determining step in which H₂O is produced as a leaving group. This is as opposed to catalysing the conversion of the carbonyl.

The clarity on the mechanism is of interest to the catalytic community particularly those investigating the liquid phase hydrogenation of alcohols. While it does appear that the conclusions are more general, and therefore of greater impact, it is less clear within the manuscript if this only applies to electrocatalytic systems. If, as suspected, it is more broadly applicable then the authors could emphasise this more within the abstract and at key points throughout.

Overall, this is an interesting manuscript and the experimental work to support the conclusions and discussion appears to be appropriate and correct. Minor points include:

Minor grammatical errors throughout e.g. (line 15) "The presence of an aqueous phase"; (line 27) "because both are more strongly coordinated" etc.

The desorption of toluene (step e) in scheme 1 (line 167) appears to be incorrectly presented. The authors are asked to check.

Scheme 2 (line 166) is incorrectly labelled as Scheme 1. Line 192 refers to Scheme 2a showing the elementary steps which appears incorrect.

Line 233 refers to the dehydration step (step e) as the RDS. However, step e appears to be the desorption of toluene. The authors are asked to check.

Line 282 (and figure 1 b) refers to the reaction order changes as a function of pH. The evidence for this would seem to be stronger when referring to figure 1 in the ESI.

Reviewer #2 (Remarks to the Author):

I have reviewed the manuscript in which the authors investigated the aqueous reduction of benzyl alcohol over a supported Pd catalyst. The researchers showed that the turnover frequency (TOF) decreased with increasing pH. By measuring the reaction order and the CV curves, the authors concluded that the surface was covered by the alcohol and its reaction intermediates while the surface H coverage was low. By measuring the kinetic isotopic effect (KIE), the authors concluded H₂ or H atoms was not involved in the rate-determining steps (RDS). They further proposed that the RDS was the

cleavage C-O bond to release a water molecule after the protonation step. This led to another hypothesis that the electric potential of the catalyst surface can stabilize the protonated species and the carbenium ion, which could lead to different reaction kinetics. This was supported by measuring TOF in an electrochemical condition where the electrode potential of Pd was tuned, and they observed TOF increased at more negative potentials. Finally, the authors tried to interpret the mechanism by analyzing the TOF as a function of the H₂ pressure and pH; they showed that increasing H₂ pressure could decrease (be more negative) electric potential, thus increasing the rate. Lowering the pH could increase population of the protonated species but also increase the potential; these two effects are opposite in determining the kinetics.

The effect of solvent and ions in solvents in heterogeneous catalysis has been an important topic. This current work shows some interesting insights on the role of electric potential of the catalyst that can be tuned by different reaction conditions (pressure, pH etc). The results were also presented in a logic way. I have some questions regarding the interpretation of the data.

1. In the KIE measurement (by measuring rates in H₂/D₂ and H₂O/D₂O), the authors concluded water or H₃O⁺ was involved in the kinetically relevant step, but later it was proposed the RDS is the C-O bond cleavage to release water. It is not clear to me how these two can be correlated. Maybe the authors should measure the KIE in O-labelled alcohol.

2. In the second scheme (by the way there are two scheme 1, and the second one should be renamed), the authors indicated the protonated species and the positively charged TS and intermediate were stabilized differently so that the intrinsic barrier changed. It is not clear to me how this was supported by the experimental results presented here.

3. In Figure 4, the authors measured the TOF as a function of electric potential; this was done by changing the electrode potential in an electrochemical measurement. It will be better if the authors can compare these electric potential to the values they obtained by changing the pH and H₂ pressure to illustrate these two were operated under similar conditions (e.g., coverage). The reason I am asking this is because the Figure 4 data was measured under a very low pH condition, which was similar to the condition in the thermal reaction at pH 0.6. In the thermal reaction the electric potential changes very minorly (maybe between about -0.05 V to -0.10 V in Figure 5) by varying the pressure at this pH. However, within this potential range, the TOF was about constant in Figure 4; this is different from a significant change in Figure 5 by varying the electric potential within a similar range.

4. As a minor question, later in the discussion, the authors talked about the pH effect on the TOF and suggested the reaction order changed when changing pH; however, Figure 1B seems to suggest the same order. This might be challenging to draw a conclusion here as the error bar was not specified, though it appears one can draw two lines in Figure 1B to indicate the different orders.

Response to reviewers' comments

Reviewers' Comments

We highly appreciate the insightful comments of the reviewers and have incorporated changes to reflect most of the suggestions and highlighted the changes within the manuscript. Here is a point-by-point response to the reviewers' comments and concerns.

Reviewer #1 (Remarks to the Author):

This manuscript provides insight into the role of hydronium ions and an open circuit potential on the hydrogenolysis of benzyl alcohol using Pd/C. This leads to the conclusion that the dehydration of the protonated alcohol is the rate-determining step in which H₂O is produced as a leaving group. This is as opposed to catalysing the conversion of the carbonyl.

The clarity on the mechanism is of interest to the catalytic community particularly those investigating the liquid phase hydrogenation of alcohols. While it does appear that the conclusions are more general, and therefore of greater impact, it is less clear within the manuscript if this only applies to electrocatalytic systems. If, as suspected, it is more broadly applicable then the authors could emphasize this more within the abstract and at key points throughout.

The conclusion drawn in the manuscript is applicable for both reactions under OCV and electrocatalytic systems. We have clarified this point in the abstract and compared the two systems in Figure 5 in the revised manuscript.

Overall, this is an interesting manuscript and the experimental work to support the conclusions and discussion appears to be appropriate and correct. Minor points include:

Minor grammatical errors throughout e.g. (line 15) "The presence of an aqueous phase"; (line 27) "because both are more strongly coordinated" etc.

The grammatical errors have been corrected accordingly in the revised manuscript.

The desorption of toluene (step e) in scheme 1 (line 167) appears to be incorrectly presented. The authors are asked to check.

The structure formula of toluene has been corrected in Scheme 1 in the revised manuscript.

Scheme 2 (line 166) is incorrectly labelled as Scheme 1. Line 192 refers to Scheme 2a showing the elementary steps which appears incorrect.

The 'Scheme 2' was wrongly labelled and we have corrected it in the revised manuscript.

Line 233 refers to the dehydration step (step e) as the RDS. However, step e appears to be the desorption of toluene. The authors are asked to check.

It should be the 'step c' rather than 'step e' as the RDS, we have corrected the sentence to "the dehydration step (**Step c**) is considered to be the RDS" in the revised manuscript.

Line 282 (and figure 1 b) refers to the reaction order changes as a function of pH. The evidence for this would seem to be stronger when referring to figure 1 in the ESI.

We meant the reaction order of hydronium ion changes during the whole pH range studied (Figure 1b), rather than the reaction order variations of benzaldehyde and H₂ changes with the pH (Figure 1 in the ESI) in Line 282. As shown in Figure 1b, which describes how the reaction rate changes as a function of the hydronium ion activity, and gives the reaction order of hydronium ions, the slope is not constant from pH 0.6 to pH 7. In order to emphasize this, we have revised Figure 1b and provide a dashed curve as a guide to the eye. A copy is shown below.

Figure 1 | Influence of pH on the activity of benzyl alcohol hydrogenolysis. (b) Log TOF as a function of pH to obtain the apparent reaction order in bulk hydronium ion with a dashed line as a guide to the eye. Reaction condition: 0.2 M buffer solution, 298 K and 1 bar H₂.

Reviewer #2 (Remarks to the Author):

I have reviewed the manuscript in which the authors investigated the aqueous reduction of benzyl alcohol over a supported Pd catalyst. The researchers showed that the turnover frequency (TOF) decreased with increasing pH. By measuring the reaction order and the CV curves, the authors concluded that the surface was covered by the alcohol and its reaction intermediates while the surface H coverage was low. By measuring the kinetic isotopic effect (KIE), the authors concluded H₂ or H atoms was not involved in the rate-determining steps (RDS). They further proposed that the RDS was the cleavage C-O bond to release a water molecule after the protonation step. This led to another hypothesis that the electric potential of the catalyst surface can stabilize the protonated species and the carbenium ion, which could lead to different reaction kinetics. This was

supported by measuring TOF in an electrochemical condition where the electrode potential of Pd was tuned, and they observed TOF increased at more negative potentials. Finally, the authors tried to interpret the mechanism by analyzing the TOF as a function of the H₂ pressure and pH; they showed that increasing H₂ pressure could decrease (be more negative) electric potential, thus increasing the rate. Lowering the pH could increase population of the protonated species but also increase the potential; these two effects are opposite in determining the kinetics.

The effect of solvent and ions in solvents in heterogeneous catalysis has been an important topic. This current work shows some interesting insights on the role of electric potential of the catalyst that can be tuned by different reaction conditions (pressure, pH etc). The results were also presented in a logic way. I have some questions regarding the interpretation of the data.

1. In the KIE measurement (by measuring rates in H₂/D₂ and H₂O/D₂O), the authors concluded water or H₃O⁺ was involved in the kinetically relevant step, but later it was proposed the RDS is the C-O bond cleavage to release water. It is not clear to me how these two can be correlated. Maybe the authors should measure the KIE in O-labelled alcohol.

According to the larger KIE for H₂O vs. D₂O compared to gaseous H₂ vs. D₂, we conclude that water and/or H₃O⁺_(hydr.) are involved in the kinetically relevant step.

The RDS step is a Brønsted acid catalyzed elimination reaction:

The rate of C-O cleavage is affected by the concentration of protonated benzyl alcohol, which has an isotope effect on whether being protonated by H⁺ or by D⁺, leading to the observed KIE in H₂O vs. D₂O.

To make it clearer, we revised the sentence in line 136 from “*This sensitivity indicates that water or H₃O⁺_(hydr.) are involved in the kinetically relevant step.*” to “*This sensitivity indicates that H or D from water or H₃O⁺_(hydr.) are involved in the kinetically relevant steps*”.

2. In the second scheme (by the way there are two scheme 1, and the second one should be renamed), the authors indicated the protonated species and the positively charged TS and intermediate were stabilized differently so that the intrinsic barrier changed. It is not clear to me how this was supported by the experimental results presented here.

The second scheme has been renamed to Scheme 2 in the revised manuscript. It schematically shows, how the electric potential on the metal surface affects the relative free energy of the initial state (C₆H₅-CH₂-OH₂⁺), corresponding transition state ([C₆H₅-

$\text{CH}_2^+\cdots\text{OH}_2$) and intermediate ($\text{C}_6\text{H}_5\text{-CH}_2^+$).

As positively charged species, they are all stabilized by more negative potential at the Pd-water interface. The relative stabilization extent depends on their location in the EDL and the distance to Pd surface because of the electric potential gradient from the surface to the bulk aqueous phase. Unfortunately, it is quite challenging to experimentally determine their location in the EDL. Nevertheless, we can still empirically estimate their relative distance to the Pd surface based on their hydrophobicity/hydrophilicity. Considering the hydrogen bonding at the outer Helmholtz layer, the benzyl carbenium ion ($\text{C}_6\text{H}_5\text{-CH}_2^+$) intermediate is assumed to be closer to the metal surface, as it has less hydrogen bonding with water compared to the initial state ($\text{C}_6\text{H}_5\text{-CH}_2\text{-OH}_2^+$). The transition state should, thus, be located in between them. Therefore, the distance to the Pd surface follows the sequence: $\text{C}_6\text{H}_5\text{-CH}_2^+ < [\text{C}_6\text{H}_5\text{-CH}_2^+\cdots\text{OH}_2] < \text{C}_6\text{H}_5\text{-CH}_2\text{-OH}_2^+$.

In consequence, the negative charge at the metal surface provides relatively more stabilization to the transition state ($[\text{C}_6\text{H}_5\text{-CH}_2^+\cdots\text{OH}_2]$) compared to the initial state ($\text{C}_6\text{H}_5\text{-CH}_2\text{-OH}_2^+$). This, consequently, decreases their energy differences and the intrinsic energy barrier. We also clarify this in the revised manuscript.

3. In Figure 4, the authors measured the TOF as a function of electric potential; this was done by changing the electrode potential in an electrochemical measurement. It will be better if the authors can compare these electric potential to the values they obtained by changing the pH and H_2 pressure to illustrate these two were operated under similar conditions (e.g., coverage). The reason I am asking this is because the Figure 4 data was measured under a very low pH condition, which was similar to the condition in the thermal reaction at pH 0.6. In the thermal reaction the electric potential changes very minorly (maybe between about -0.05 V to -0.10 V in Figure 5) by varying the pressure at this pH. However, within this potential range, the TOF was about constant in Figure 4; this is different from a significant change in Figure 5 by varying the electric potential within a similar range.

We compared the reaction rates obtained both at the open circuit potential and external electric potentials as a function of the electric potential in the revised Figure 5. To make those data available to compare, we used Nernst equation to calculate a hypothetical equilibrated H_2 pressure under each tested external potential. A copy of Figure 5 is shown below. For each pH, the reaction rate increases with H_2 pressure, caused by the decreasing of electric potential at either OCP or under external electric potentials. In comparison, for a certain electric potential, the reaction rate increases with decreasing

pH, which is ascribed to the increase in BAH^+_{ad} coverage.

Figure 2 | (a) TOF as a function of electric potential of Pd (ϕ_M) and P_{H_2} at OCP under pH 0.6 (0.5-20 bar), pH 2.5 (0.3-10 bar) and pH 5 (0.2-10 bar) and external electric potential at pH 1.6 on Pd/C (5wt.%) at 298K. **(b)** The corresponding projection onto the plane of TOF vs. ϕ_M . A hypothetical P_{H_2} is used for pH 1.6 that is equilibrated with the externally added electric potentials.

4. As a minor question, later in the discussion, the authors talked about the pH effect on the TOF and suggested the reaction order changed when changing pH; however, Figure 1B seems to suggest the same order. This might be challenging to draw a conclusion here as the error bar was not specified, though it appears one can draw two lines in Figure 1B to indicate the different orders.

In the previous manuscript, we gave a straight eye-guiding line in Figure 1b. However, based on the experimental data and our proposed reaction mechanism, it should be a curve. We have changed it in the revised manuscript and a copy of Figure 1b is shown below. With increasing pH, the slope of $\log\text{TOF}$ vs. pH becomes more negative, i.e., the reaction order in hydronium ion becomes larger. This is due to the smaller coverage of BAH^+_{ad} at a higher pH, which is in accordance with the relation below.

$$\frac{\partial \ln r}{\partial \ln a_{\text{H}^+}} = 1 - \theta_{\text{BAH}^+_{\text{ad}}} - F \frac{\partial \varphi_{\text{TS}}}{\partial \mu_{\text{H}^+}} + \theta_{\text{BAH}^+_{\text{ad}}} F \frac{\partial \varphi_{\text{BAH}^+_{\text{ad}}}}{\partial \mu_{\text{H}^+}}$$

Figure 3 | Influence of pH on the activity of benzyl alcohol hydrogenolysis. (b) Log TOF as a function of pH to obtain the apparent reaction order in bulk hydronium ion with a dashed line as a guide to the eye. Reaction condition: 0.2 M buffer solution, 298 K and 1 bar H₂.

REVIEWERS' COMMENTS

Reviewer #2 (Remarks to the Author):

I have reviewed the revised manuscript and find that my comments have been addressed by the authors properly. A couple of the figures have been revised and more clarification has been included. I recommend its publication in the current format.

Response to Reviewers' comments.

We are grateful to the reviewers' comments and suggestions to help us improve the manuscript.

Reviewer #2 (Remarks to the Author):

I have reviewed the revised manuscript and find that my comments have been addressed by the authors properly. A couple of the figures have been revised and more clarification has been included. I recommend its publication in the current format.

We thank Reviewer 2 for the highly remarks.